# Comparison between enzyme-linked immunospot assay and intracellular cytokine flow cytometry assay of cytomegalovirus-specific T-cell response in healthy participants

Chompunuch Klinmalai[1], Nopporn Apiwattanakul[1], Somsak Prasongtanakij[2], Jackrapong Bruminhent[3,4], Supanart Srisala[2]*

1 Department of Paediatrics, Faculty of Medicine Ramathibodi Hospital, Mahidol University, Bangkok, Thailand, 2 Research Laboratory Section, Office of Health Science Research, Faculty of Medicine Ramathibodi Hospital, Mahidol University, Bangkok, Thailand, 3 Division of Infectious Diseases, Department of Medicine, Faculty of Medicine Ramathibodi Hospital, Mahidol University, Bangkok, Thailand, 4 Ramathibodi Excellence Center for Organ Transplantation, Faculty of Medicine Ramathibodi Hospital, Mahidol University, Bangkok, Thailand

* supanart.sri@mahidol.ac.th

## Abstract

Human cytomegalovirus (HCMV) usually establishes a lifelong latent infection after primary infection. Reactivation occurs sporadically and is controlled by cell-mediated immune response (CMIR). Monitoring of CMIR against CMV is mandatory in immunocompromised patients to adjust immunosuppressive drugs to prevent serious end-organ damage after CMV reactivation. Intracellular staining (ICS) and enzyme-linked immunospot (ELISPOT) quantifying CMV-specific T-cells are generally used as surrogate markers for CMIR against CMV. Whether the results of these 2 methods correlate well is not known. This study compared the numbers of CMV-specific T-cells identified by ELISPOT and ICS in healthy adult volunteers. Correlation with CMV serological status was explored. Thirty peripheral blood samples from healthy individuals were quantified for IFN-γ-producing cells after stimulation with whole CMV and IE1 using ICS, and IFN-γ-secreting cells after stimulation with whole CMV lysate and IE1 peptide pool using ICS and IFN-γ-secreting cells by ELISPOT. Anti-CMV IgG levels were analyzed concomitantly using a chemiluminescent microparticle immunoassay. There were 30 healthy participants, 15 (50%) male, with a mean age of 37.8 (± 7.6) years. Twenty-eight (93.3%) were seropositive against CMV. The CMV-specific CD3+ cells, as measured by ICS, were highly correlated with the spot numbers obtained by the ELISPOT, irrespective of CMV antigens used (whole CMV, r = 0.7677, p < 0.0001; IE1, r = 0.6516, p < 0.0001). The numbers of CMV-specific CD3+ cells quantified by IE1 stimulation by these 2 assays were statistically correlated with anti-CMV IgG levels (ICS, r = 0.5070, p = 0.004; ELISPOT, r = 0.4384, p = 0.015). Among the 30 participants, CMV-specific T cells were detected in all participants (100%), including the two seronegative individuals. The present

**Data availability statement:** All relevant data are within the paper and its Supporting information files.

**Funding:** Faculty of Medicine, Ramathibodi Hospital, Mahidol University, Thailand, Grant Number: RF_66114. The funders had no role in study design, data collection and analysis, decision to publish, or preparation of the manuscript.

**Competing interests:** The authors have declared that no competing interests exist.

study demonstrated that CMV-specific T-cells measured by ICS and ELISPOT assays were well correlated, suggesting that these assays could be used to monitor CMV-specific T-cells. CMV IgG levels may reflect prior CMV infection and CMV-specific cell-mediated immunity (CMIR) in immunocompetent individuals.

## Introduction

Human cytomegalovirus (HCMV) is a member of the Betaherpesvirinae. Like all herpesviruses, primary infection is rapidly controlled by the immune system, but the virus establishes a lifelong latent infection. CMV reactivation typically remains asymptomatic in immunocompetent hosts, as robust immune surveillance prevents clinical disease. However, this can lead to uncontrolled CMV replication and severe end-organ damage such as retinitis, pneumonitis, colitis, and encephalitis in immuno-compromised patients [1,2]. Cell-mediated immune response (CMIR) plays a crucial role in controlling CMV replication. CD4$^+$T helper and CD8$^+$cytotoxic T lymphocytes are key mediators of CMV-specific immunity. Interferon-gamma (IFN-γ) production following stimulation with CMV antigens is commonly used as a functional readout of CMV-specific T-cell responses [3–6]. IFN-γ plays an important role in activating both macrophages and T-cells, leading to the subsequent promotion of innate and adaptive immune response and, subsequently, prevention of CMV replication [7]. In addition to T lymphocytes, other immune cells also contribute to host defense against CMV infection. In particular, natural killer (NK) cells and natural killer T (NKT) cells have been reported to participate in the early immune response against CMV through cytotoxic activity and cytokine production [8–10]. CMIR monitoring is therefore essential in immunocompromised patients vulnerable to CMV infection or reactivation such as organ transplant patients. This requires quantifying CMV-specific T-cells, which respond *in vitro* to CMV antigens. CMV-specific T-cell quantification is mainly based on re-stimulating CD4$^+$ and/or CD8$^+$ effector cells with CMV antigens or pep-tide pools. If CMV-specific T-cells are present, cytokines will be produced after cell stimulation. These cytokines include but are not limited to interferon-gamma (IFN-γ), tumor necrotic factor-alpha (TNF-α), granzyme B which could be measured by several techniques such as cytokine secretion assay (CSA), cytokine-specific enzyme-linked immunosorbent assay (ELISA), intracellular cytokine staining assay (ICS) using flow cytometry, and enzyme-linked immunospot assay (ELISPOT). This could identify immunocompromised individuals at risk of CMV reactivation or disease [11,12].

ELISPOT-based assay represents the most sensitive read-out system. It thus is suitable for detecting low-level responses from isolated peripheral blood mononuclear cells at the single-cell level both qualitatively and quantitatively [13,14]. However, variation in cut-off values from many studies led to an inconclusive reference range to determine whether patients had adequate CMIR against CMV [15–17]. In addition, ELISPOT-based assay cannot determine the exact cell sources of cytokine production. In contrast, ICS is more technically laborious but can provide the cell sources of cytokine production [18].

Although cell-mediated immunity is considered the primary mechanism for controlling CMV infection, humoral immune responses may contribute to limiting viral dissemination in certain contexts, particularly by reducing viral spread and reinfection, but are not sufficient on their own [19]. Humoral immune response (HIR) could be crucial in restricting CMV dissemination and minimizing disease severity [20]. Anti-CMV IgM could be found in primary infection and tends to disappear 3–6 months after the primary infection but can re-appear when re-infection by a different viral strain or reactivation occurs [21]. Notably, while anti-CMV IgG antibody levels correlate with the frequency of circulating CMV-specific memory B-cells [22] and serve as a reliable indicator of prior exposure, they do not directly reflect the functional vigor of the CMIR. Consequently, serological status alone is insufficient to assess comprehensive immunological protection, as T-cell-mediated pathways remain the cornerstone for controlling viral latency and preventing clinical reactivation [1,23].

This study compared ELISPOT and intracellular cytokine staining (ICS) flow cytometry assays for the quantification of CMV-specific T-cell responses in peripheral blood mononuclear cells (PBMCs) from healthy individuals following CMV antigen stimulation. In addition, the study explored the association between anti-CMV IgG levels and CMV-specific cell-mediated immune responses (CMIR).

## Methods

### Participants

Participants aged greater than 18 years old were included between November 7th, 2023, and March 31th, 2024. Those who had received immunosuppressive drugs or had immunocompromised conditions were excluded. This study protocol was approved by the Research Ethical Committee of Ramathibodi Hospital, Mahidol University (COA. MURA2023/418) on May 22nd, 2023, with renewal approved on May 22nd, 2024 (MURA2023/418 Ref.2024/439). Participants provided written informed consent.

### Blood collection and PBMC preparation

Blood samples were collected in heparinized tubes by venipuncture and stored for up to 3 hours at room temperature (18-25°C) until further processing. Peripheral blood mononuclear cells (PBMC) were freshly separated from whole blood by Lymphoprep™ (STEMCELLS, Germany) density gradient centrifugation. The cells were then thoroughly washed with phosphate-buffered saline (PBS) (Sigma Aldrich, Germany) and re-suspended in RPMI medium (Gibco; Gaithersburg, MD, USA) supplemented with 10% fetal bovine serum (Gibco; Gaithersburg, MD, USA).

### ELISPOT assay

IFN-γ secreted by activated PBMCs was detected using ELISPOT assays using a Human IFN-γ ELISPOT PRO kit (ALP) (Mabtech Stockholm, Sweden) according to the manufacturer's protocol. The ELISPOT plates were washed 5 times with 200 µL/well of PBS. After 5 times of washing, 200 µL/well of RPMI-1640 media with 10% fetal bovine serum was added, and the plates were incubated for 30 minutes at room temperature. The plates were washed and PBMCs ($2.5 \times 10^5$ cells per 200 µL) re-suspended in RPMI medium supplemented with 10% fetal bovine serum were cultivated in each well of the plates. Cells re-suspended in RPMI medium were served as a negative control. Cells incubated with anti-CD3 antibody were served as a positive control. Tested cells were incubated with either PepMix™ CMV IE1 (JPT Peptide Technologies (Berlin, Germany)) (1 µg/mL) or whole-CMV lysate (wCMV) (Abcam Biotechnology company (Cambridge, UK)) (5 µg/mL). After incubation for 40 hours at 37°C with a 5% $CO_2$ supplement, the cells were removed, and IFN-γ was determined by incubating the plates with monoclonal antibody (7-B6-1-ALP) for 2 hours at room temperature, followed by treatment with 100 µL of ready-to-use BCIP®/NBT-plus substrate. The plates were washed 5 times with 200 µL/well of Dulbecco's PBS. The spots were then counted with an Immunospot analyzer (Cellular Technology Limited), and spot quality was checked using Immunospot software version 7.0.26.0. Results were reported as IFN-γ producing spot-forming units (SFUs) per $10^6$ PBMCs.

### Detection of CMV-specific T-cells by intracellular cytokine staining (ICS)

PBMC preparation was performed as previously described [24]. PBMCs suspended in RPMI-1640 medium containing 10% FBS were stimulated with whole-CMV lysate (wCMV) at a concentration of 5 µg/ml or PepMix™ CMV (IE1) at 1 µg/ml for 18 hours at 37°C supplemented with 5% $CO_2$ in humidified air. Brefeldin A was added to a final 1 mg/ml concentration during the last 3 hours of incubation. Cells treated with 12.5 ng/mL of phorbol 12-myristate 13-acetate (PMA) (Sigma Aldrich, Inc.; St. Louis, MO, USA) and 2 µg/ml of ionomycin (Sigma Aldrich, Inc.; St. Louis, MO, USA) were used as a positive control. Untreated cells were used as a negative control. After cultivation, the harvested cells were fixed in 1% formaldehyde (Sigma-Aldrich, Inc.; St. Louis, MO, USA) for 15 minutes at room temperature. The cells were then washed with PBS, re-suspended in 0.1% saponin (Sigma-Aldrich, Inc.; St. Louis, MO, USA), and incubated at room temperature for 15 minutes. After washing, they were stained with fluorescence-tagged antibodies suspended in 0.1% saponin. The antibody cocktail consists of 1:250 dilutions of anti-CD3 FITC (clone HIT3a), anti-CD4 APC (clone OKT4), anti-CD8a APC/EF780 (clone RPA-T8), anti-CD56 PE (clone CMSSB), and anti-IFN-γ PE-Cyanine7 (clone 4SB3). All antibodies were purchased from ThermoFisher Scientific (San Diego, CA, USA). Cell staining was performed at 4°C in the dark for 30 minutes, after which they were subjected to flow cytometry analysis using FACSVerse Flow cytometer (BD Biosciences, USA). All flow cytometry analyses were performed using FlowJo® software version 10.8.0 (FlowJo LLC, USA). Supporting information: S1 Fig. summarizes each cell type's gating strategy. Only the percentages of lymphocyte subsets with regard to total lymphocyte count will be presented. Percentages of CMV-specific T-cells of each T-cell subset were expressed as those T-cells producing IFN-γ per $10^6$ cells of each T-cell subset.

### Humoral immune responses

Sera were collected by centrifugation of clot blood at 2566 g for 10 minutes at room temperature and were stored at −80 °C before anti-CMV IgG level analysis. Anti-CMV IgG levels were measured by chemiluminescent microparticle immunoassay. Quantitative anti-CMV IgG results are reported in AU/ml; seropositive was defined as anti-RBD IgG levels ≥ 6.6 AU/ml (Alinity i CMV IgG Assay – Cytomegalovirus (CMV) immunoglobulin G (IgG) antibody IVD kit, Abbott, Abbott Park, Illinois, U.S.A.).

### Statistical analysis

Statistical analyses were performed using STATA (StataCorp, version 17, College Station, TX, USA). The graph plots were conducted using GraphPad Prism version 9.0 (GraphPad, Inc., San Diego, CA, USA). A comparison of the cytomegalovirus (CMV) antigens stimulated group, and the non-stimulated group was performed using the Wilcoxon signed-rank test. The strength of the correlation between the flow-cytometry ICS assay and the ELISPOT assay, the correlation between ICS or ELISPOT assays and anti-CMV IgG were analyzed using Spearman's rank correlation. The corresponding 95% confidence intervals (95% CI) for correlation coefficients were calculated to assess the precision of the estimated correlations. Agreement between the ICS and ELISPOT assays was further evaluated using Bland–Altman analysis, which was used to estimate the mean bias and the 95% limits of agreement between the two methods. Anti-CMV IgG level > 250 AU/mL was treated as the level of 250 AU/mL in the correlation analysis. $P$-value < 0.05 was considered to be statistically significant.

## Results

### Demographic data

The demographic data of the participants are shown in Table 1. Among 30 participants, 15 (50%) were male. Their ages ranged from 19 to 59 years, with a mean value of 37.8 years. The average body mass index (BMI) was 23.09 Kg/m² (range 16.7–33.8 Kg/m²). All participants had no history of immune diseases, active viral infections, or other underlying diseases.

**Table 1. Demographic characteristics of the study participants.**

| Characteristics | Number (%) or mean (SD) |
| --- | --- |
| Healthy individual number (n) | 30 |
| Age (years) | 37.8 (7.6) |
| Gender, n (%) | |
| • Male | 15 (50%) |
| • Female | 15 (50%) |
| BMI | 23.09 (3.85) |

Abbreviations: SD, Standard deviation.

## Evaluation of CMV-specific T-cells

The two assays, ICS by flow cytometry with the quantification of IFN-γ producing cells (Fig 1) and enzyme-linked immuno-spot (ELISPOT) assays with the quantification of IFN-γ secreting T-cells (Fig 2), were conducted in parallel.

The number of IFN-γ producing cells is expressed per $10^6$ cells of each T-cell subset and is shown in Table 2. There is a significant difference in the median of IFN-γ producing cells in all lymphocyte subsets (CD4$^+$, CD8$^+$, and NK cells) between negative control and cells stimulated with whole CMV lysate or IE1 peptide ($p < 0.0001$ for every cell type). The number of IFN-γ-secreting spot-forming units (SFUs) per $10^6$ PBMCs in each subject is provided in Table 3.

## Comparisons of the two cellular assays

The CMV-specific IFN-γ producing T-cell responses in samples from 30 healthy subjects measured by ICS were compared with those measured by ELISPOT assays. Spearman's rank correlation analyses revealed that the T-cell response from ICS was significantly correlated with that from the ELISPOT assay, irrespective of CMV antigens (wCMV, IE1) (Fig 3). The CMV-specific CD3$^+$ cell responses measured by ICS showed a highly positive correlation with the spot number derived from ELISPOT assay (wCMV, r = 0.7677, $p < 0.0001$, 95% CI [0.5556, 0.8860]; IE1, r = 0.6516, $p < 0.0001$, 95% CI [0.3711, 0.8231]) (Fig 3A and 3B). Relatively weak correlations were observed between the CMV-specific CD4$^+$ T-cell responses stimulated by IE1 and the spot number derived from ELISPOT (r = 0.3726, $p = 0.0426$, 95% CI [0.0031, 0.6526]) (Fig 3D). Moreover, the CMV-specific CD8$^+$ T-cell responses measured by ICS showed a moderate positive correlation with the spot number derived from ELISPOT (wCMV, r = 0.5617, $p = 0.0012$, 95% CI [0.2421, 0.7714]; IE1, r = 0.4715, $p = 0.0085$, 95% CI [0.1230, 0.7164]) (Fig 3E and 3F). However, the CMV-specific NK-cell response measured by ICS showed no significant correlation with the total spot number of ELISPOT, irrespective of CMV antigens (wCMV, r = 0.1634, $p = 0.3883$, 95% CI [-0.2198, 0.5029]; IE1, r = 0.2403, $p = 0.2008$, 95% CI [-0.1423, 0.5604]) (Fig 3I and 3J). The Wilcoxon signed-rank test was also used to compare the frequencies of CMV-specific T-cells measured by ICS and ELISPOT assays, and no significant difference between the results of the two methods in CD3$^+$ cell responses could be demonstrated, irrespective of CMV antigens (wCMV, $p = 0.7303$; IE1, $p = 0.2534$).

## Correlation of humoral and cellular immune response against CMV

Correlation between the numbers of CMV-specific T-cells derived from ICS or ELISPOT assays and anti-CMV IgG level was tested. Statistically significant correlations between CMV-specific IFN-γ-producing CD3$^+$, CD8$^+$, and double negative T-cells after being stimulated with IE1 and quantitative anti-CMV IgG level could be demonstrated (Table 4). The numbers of cells secreting IFN-γ after being stimulated by IE1 measured by ELISPOT also showed a statistically significant correlation with quantitative anti-CMV IgG level. However, there is no correlation between IFN-γ-producing cells detected by stimulation with wCMV either by ICS or ELISPOT assays and quantitative anti-CMV IgG level. Interestingly, 2 of the 30

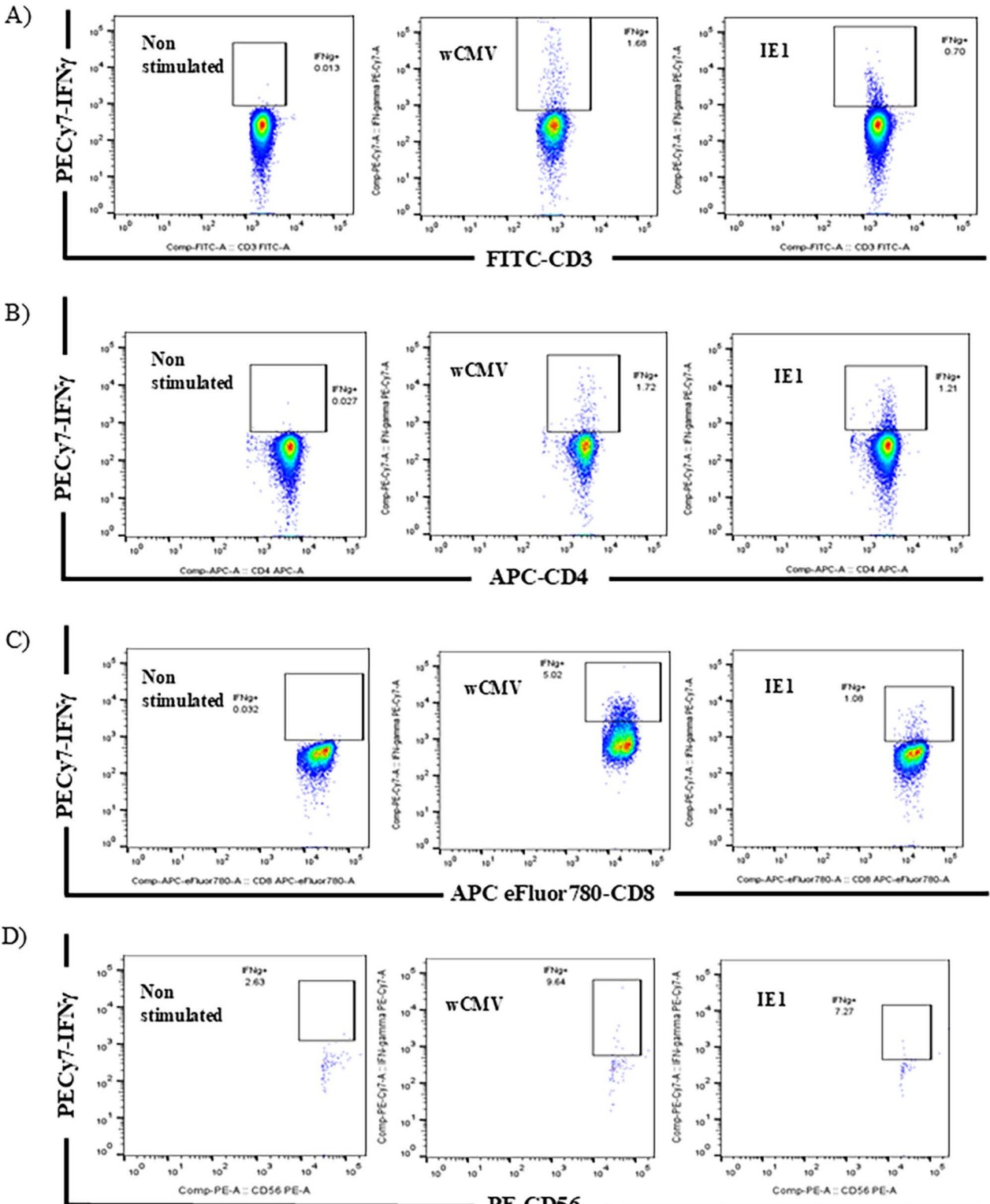

**Fig 1. Demonstration of CMV-specific T-cells and NK-cells.** Peripheral blood mononuclear cells (PBMC) were stimulated overnight with whole CMV lysate or IE1 antigen, and intracellular staining of IFN-γ was performed. Gated from the CD3+ lymphocyte population (A), both CD4+ T-cells (B) and CD8+

T-cells (C) released IFN-γ upon the stimulation with wCMV or IE1. When the lymphocyte population was gated with CD56⁺ (a marker of NK-cells), some of these cells could release IFN-γ upon stimulation with the antigens (D).

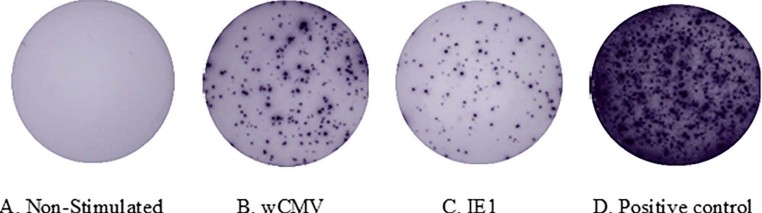

A. Non-Stimulated B. wCMV C. IE1 D. Positive control

**Fig 2. Detection of CMV-specific IFN-γ secreting cells by Enzyme-linked immunospot assay (ELISPOT) assay.** ELISPOT images of cells for negative control (A), stimulated with wCMV lysate (B), IE1 peptide (C), and for positive control (D).

**Table 2. Immunological parameters of enrolled subjects.**

| Parameters | Median (IQR) |
|---|---|
| Lymphocyte subsets | |
| • CD3⁺ (% total lymphocyte) | 72.9 (68.4, 76.5) |
| • CD4⁺ (% total lymphocyte) | 38.6 (35.9, 47.8) |
| • CD8⁺ (% total lymphocyte) | 25.0 (19.7, 30.9) |
| • Double negative (% total lymphocyte) | 3.8 (2.8, 5.9) |
| • NK (% total lymphocyte) | 0.2 (0.1, 0.3) |
| IFN-γ producing cells stimulated by IE1 | |
| • CD3⁺ (/10⁶ CD3⁺cells) | 1,083.1 (552.6, 338.5) |
| • CD4⁺ (/10⁶ CD4⁺cells) | 1,752.1 (543.2, 5,434.6) |
| • CD8⁺ (/10⁶ CD8⁺cells) | 2,902.2 (1,416.4, 5,206.8) |
| • Double negative (/10⁶ Double negative cells) | 6,943.4 (2,625.3, 10,269.2) |
| • NK (/10⁶ NK cells) | 42,003.7 (21,088, 79,685.4) |
| IFN-γ producing cells stimulated by wCMV | |
| • CD3⁺ (/10⁶ CD3⁺cells) | 1,856.2 (991.4, 4,498.3) |
| • CD4⁺ (/10⁶ CD4⁺cells) | 2,358.9 (1,375.3, 5,316.9) |
| • CD8⁺ (/10⁶ CD8⁺cells) | 4,248.1 (1,549.6, 8883.3) |
| • Double negative (/10⁶ Double negative cells) | 8,865.9 (4,263.9, 17,976.6) |
| • NK (/10⁶ NK cells) | 50,897.5 (19,374.1, 106,060.6) |
| ELISPOT stimulated by IE1 (/10⁶ PBMC) | 1,597.5 (955, 2,435) |
| ELISPOT stimulated by wCMV (/10⁶ PBMC) | 2,377.5 (1,480, 3,820) |
| CMV IgG (AU/mL) | 148.4 (105.4, 245.5) |

IQR: interquartile range; Double negative: double negative T-cell (CD3+CD4-CD8-); NK: natural killer cell; IFN-γ: interferon gamma; IE1: immediate early protein 1 peptide pool; wCMV: whole cell CMV lysate; PBMC: peripheral blood mononuclear cell; AU/mL: arbitrary unit per milliliter.

**Table 3. IFN-γ ELISPOT assay of individual participants.**

| Subject Number | IFN-γ secreting spot-forming units (SFUs) per $10^6$ PBMCs | | |
| --- | --- | --- | --- |
| | Non-stimulated cells | wCMV-stimulated cells | IE1-stimulated cells |
| 1 | 125 | 1960 | 630 |
| 2 | 430 | 2300 | 675 |
| 3 | 240 | 350 | 550 |
| 4 | 480 | 1960 | 5320 |
| 5 | 50 | 3245 | 1460 |
| 6 | 525 | 9920 | 5515 |
| 7 | 335 | 4155 | 5005 |
| 8 | 280 | 4220 | 1130 |
| 9 | 2495 | 3220 | 3945 |
| 10 | 490 | 4400 | 6250 |
| 11 | 310 | 1405 | 1265 |
| 12 | 2275 | 3590 | 3265 |
| 13 | 1145 | 4285 | 2620 |
| 14 | 410 | 1345 | 2420 |
| 15 | 4010 | 5015 | 4735 |
| 16 | 50 | 4160 | 1740 |
| 17 | 615 | 3160 | 2505 |
| 18 | 295 | 2170 | 2670 |
| 19 | 310 | 1795 | 2145 |
| 20 | 115 | 6500 | 4610 |
| 21 | 315 | 2525 | 1980 |
| 22 | 670 | 2630 | 4185 |
| 23 | 455 | 3590 | 2505 |
| 24 | 0 | 6110 | 270 |
| 25 | 380 | 3705 | 1800 |
| 26 | 320 | 3880 | 1830 |
| 27 | 35 | 9550 | 3010 |
| 28 | 40 | 1845 | 1570 |
| 29 | 15 | 2875 | 2450 |
| 30 | 0 | 5 | 95 |

participants (Fig 4A and 4B) had seronegative levels of anti-CMV IgG (anti-RBD IgG levels < 6.6 AU/ml) but had CMV-specific T-cells detected by both the ICS and ELISPOT assays.

**Bland–Altman analysis**

To further evaluate the agreement between intracellular cytokine staining and ELISPOT assay, Bland–Altman analysis was performed. For stimulation with whole CMV lysate, the mean bias was 2,599 cells, with 95% limits of agreement from −13,338–18,535 cells. For stimulation with IE1, the mean bias between the two assays was 237 cells per $10^6$ CD3$^+$T cells, with 95% limits of agreement ranging from −3,676–4,151 cells. It should be noted that the results of the ELISPOT assay were expressed as spots per $10^6$ peripheral blood mononuclear cells (PBMCs), whereas ICS results were reported as cytokine-producing cells per $10^6$ CD3$^+$T cells (Fig 5).

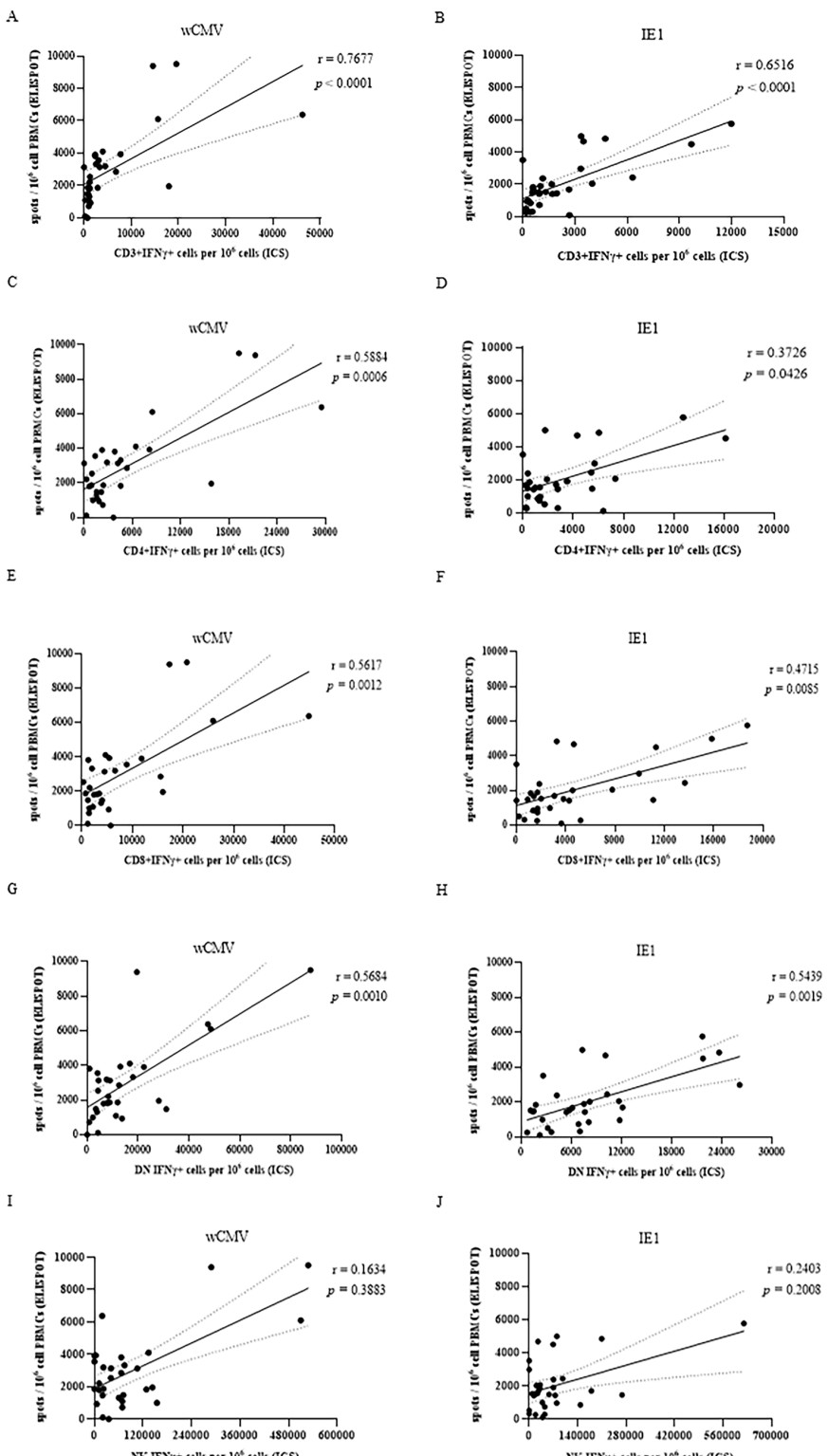

**Fig 3. Correlations of CMV-specific IFN-γ producing/secreting cells per 10$^6$ cells between intracellular staining by flow cytometry (ICS) and enzyme-linked immunospot (ELISPOT) assay.** The correlation coefficient was calculated with a Spearman's rank test, and *p* values <0.05 were considered statistical significance. Upper and lower dot lines represent 95% confidence intervals of correlation coefficients.

**Table 4. Correlation between IFN-γ producing/secreting cells and quantitative CMV IgG.**

| IFN-γ producing cells | Spearman's rho | 95% CI | *p*-value |
|---|---|---|---|
| CD3⁺ stimulated by IE1 | 0.5070 | [0.1716, 0.8424] | 0.004 |
| CD3⁺ stimulated by wCMV | 0.0601 | [-0.3175, 0.4377] | 0.753 |
| CD4⁺ stimulated by IE1 | 0.3221 | [-0.0675, 0.7116] | 0.083 |
| CD4⁺ stimulated by wCMV | −0.0639 | [-0.4254, 0.2977] | 0.737 |
| CD8⁺ stimulated by IE1 | 0.4038 | [0.0519, 0.7606] | 0.027 |
| CD8⁺ stimulated by wCMV | 0.1356 | [-0.2361, 0.5073] | 0.475 |
| Double negative stimulated by IE1 | 0.5436 | [0.2345, 0.8527] | 0.002 |
| Double negative stimulated by wCMV | 0.2055 | [-0.2066, 0.6176] | 0.276 |
| NK stimulated by IE1 | 0.2689 | [-0.1076, 0.6453] | 0.151 |
| NK stimulated by wCMV | −0.0793 | [-0.4364, 0.2777] | 0.677 |
| ELISPOT stimulated by IE1 | 0.4384 | [0.0632, 0.8137] | 0.015 |
| ELISPOT stimulated by wCMV | 0.2200 | [-0.1682, 0.6082] | 0.243 |

Double negative: double negative T-cell (CD3＋CD4-CD8-); NK: natural killer cell; IFN-γ: interferon gamma; IE1: immediate early protein 1 peptide pool; wCMV: whole cell CMV lysate.

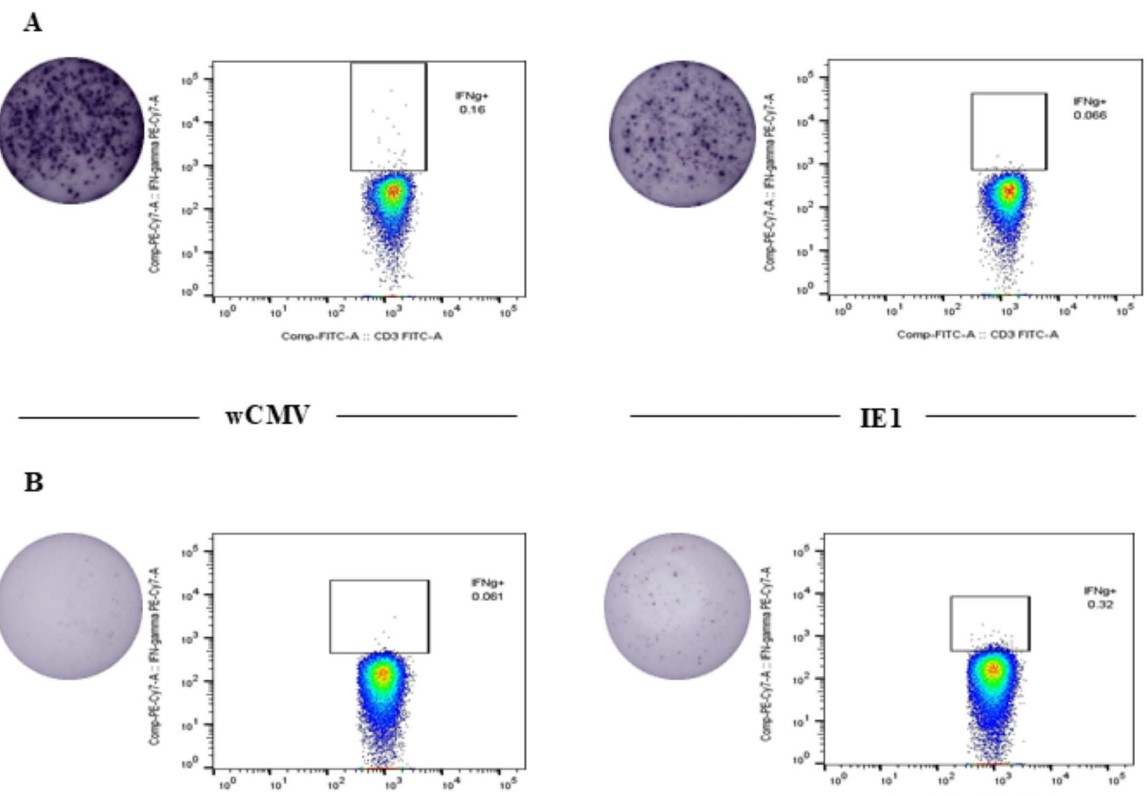

**Fig 4. Flow cytometry and corresponding ELISPOT images of two negative serum samples.**

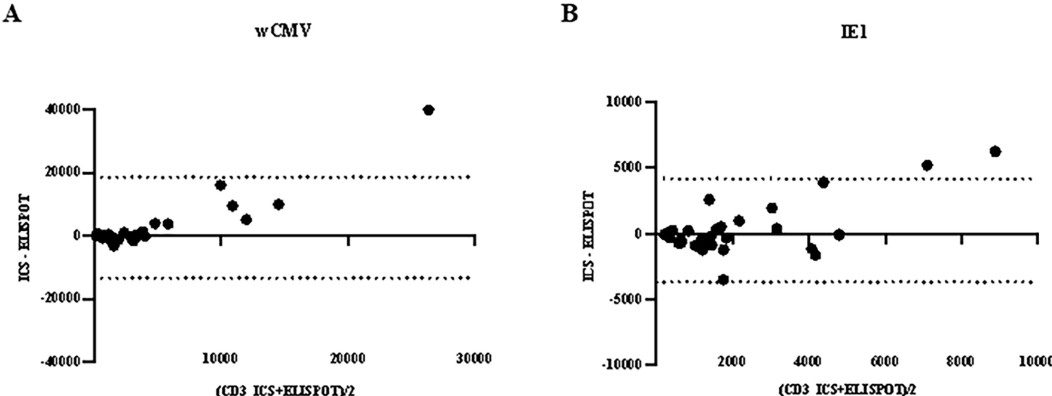

**Fig 5. Bland–Altman plots for agreement between the enzyme-linked immunospot (ELISPOT) assay and intracellular staining by flow cytometry (ICS).** Bland–Altman analyses of spots per $10^6$ peripheral blood mononuclear cells (PBMCs) and cytokine-producing cells per $10^6$ CD3$^+$T cells were performed; stimulated with wCMV lysate (A) and IE1 peptide (B). Middle lines represent mean bias values. Upper and lower dot lines represent 95% confidence intervals of upper and lower limits of agreement, respectively.

## Discussion

Despite previous reports describing correlations between ELISPOT assay and Intracellular Cytokine Staining for the detection of CMV-specific T-cell responses to Cytomegalovirus, comparative data evaluating these two platforms together with CMV-specific humoral immunity remain limited. In particular, the relationship between CMV-specific T-cell responses measured by these assays and anti-CMV IgG levels has not been fully characterized in healthy individuals. Clarifying this relationship may help improve the understanding of the complementary roles of cellular and humoral immune responses and provide practical insights into assay selection for CMV immune monitoring. Therefore, this study aimed to provide additional analytical validation of the relationship between ELISPOT and ICS assays in a cohort of healthy adults and to further explore their association with CMV-specific antibody levels.

Intracellular cytokine staining (ICS) and enzyme-linked immunospot (ELISPOT) assays have been widely used to quantify CMV-specific T-cells. Several previous studies have already demonstrated a significant correlation between these two assays; however, direct comparative evaluations under identical experimental conditions remain limited. The results of the present study showed that CMV-specific T-cells measured by ICS and ELISPOT assays showed a significant correlation. Furthermore, a stronger correlation was observed with whole CMV stimulation than with CMV IE1 peptide pool stimulation. However, the CMV-specific NK-cells measured by ICS showed no significant correlation with the spot number of ELISPOT, irrespective of CMV antigens.

While previous studies have reported correlations between ELISPOT and ICS assays, our findings provide additional analytical validation of their relationship in a cohort of healthy adults and extend prior observations by exploring the association between CMV-specific cellular responses and CMV-specific humoral immunity. Previous studies have demonstrated that CMV-specific CD8$^+$ T-cell response in patients with hypertension measured by ICS and ELISPOT assays had good agreement and significant correlation with each other [25,26], supporting the results of the present study. The precision is comparable between ICS and ELISPOT [27].

The Bland–Altman analysis demonstrated relatively small mean biases between the two assays; however, the wide limits of agreement indicate considerable variability between measurements obtained by the two methods. One possible explanation for this variability is the difference in the denominators used for quantification. The ELISPOT assay reported responses as spots per $10^6$ peripheral blood mononuclear cells, whereas ICS quantifies cytokine-producing cells per $10^6$ CD3$^+$T-cells. Although both assays showed significant correlations and similar overall trends in detecting CMV-specific T-cell responses to

Cytomegalovirus antigens, the relatively wide limits of agreement suggest that ELISPOT and ICS should not be considered directly interchangeable for quantitative measurements. Instead, these assays may provide complementary information for assessing CMV-specific cellular immune responses. The different characteristics of these assays signify the appropriate techniques selected to use in specific laboratory settings. From a practical perspective, ELISPOT offers a relatively simple and sensitive platform for detecting antigen-specific T-cell responses, whereas ICS allows detailed phenotypic characterization of responding T-cell subsets. The correlation observed between these two assays suggests that ELISPOT may serve as a practical alternative for CMV immune monitoring in laboratories where flow cytometry-based ICS is not routinely available. Notably, several features of the protocols could cause a discrepancy in the responses between these two methods. ICS can determine the number of CMV-specific T-cell subsets because specific cell surface markers can be specifically stained during flow cytometry. The ELISPOT-based method quantifies the frequency of all cytokine-secreting cells, but the lymphocyte subsets responsible for cytokine production cannot be identified. However, ELISPOT was typically favored for its low detection threshold [25] because ELISPOT assay could be performed with relatively low cell numbers. Nevertheless, ICS preferred for its greater sensitivity in detecting cells that secrete low levels of IFN-γ [28]. This study demonstrated a good correlation between IFN-γ-producing cells quantified by ELISPOT and ICS, especially between IFN-γ-producing CD3$^+$ and ELISPOT, indicating that the IFN-γ-producing cells within the PBMC are primarily CD3$^+$ cells, as previously reported [25]. These findings further highlight the complementary roles of ELISPOT and ICS assays in CMV immune monitoring, depending on whether a laboratory prioritizes assay sensitivity or detailed cellular phenotyping. The lack of correlation between IFN-γ-producing NK-cells and ELISPOT may reflect a low percentage of NK-cells in PBMC. The ability to discriminate between CD4$^+$ and CD8$^+$ responses could be beneficial in some circumstances. Long-term protection from CMV infection may be better reliable on the CD4$^+$ T-cell response, rather than the CD8$^+$ T-cell response, is restored [11]. In fact, CMV-specific CD8$^+$ T-cells may not function well in the absence of the CD4$^+$ T-cell counterpart [29,30]. Another limitation of the ELISPOT assay is that if the cell spots are more than 1,000 spots/well, the counting accuracy would be compromised [25].

Another point to be addressed is the correlations between humoral and cellular immune responses against CMV infection in healthy subjects. Correlations between quantitative anti-CMV IgG level and numbers of IFN-γ-producing cells after being stimulated by IE1, but not whole-cell CMV lysate, could be demonstrated. Of note, the correlations were significant in CD3$^+$, CD8$^+$, and double-negative T-cells and nearly significant in CD4$^+$ T-cells. This implies that the humoral immune response to CMV infection may better parallel numbers of CMV-specific CD8$^+$ and double-negative T-cells, which are considered cytotoxic T-cells. Double-negative T-cells could act as cytotoxic T-cells because they can elicit cytotoxic activity to viral and bacterial infections [31,32]. Therefore, anti-CMV IgG level indicating past or recent CMV infection would also represent the magnitude of cellular immune response mediated by CD8$^+$ or double-negative T-cells in healthy volunteers. Interestingly, among the two seronegative anti-CMV IgG participants, both the ICS and ELISPOT assays could identify CMV-specific T-cells, and the number of cells could be significantly high. These findings suggest that CMV-specific T-cells can be functional even without humoral immune response, especially in the early stage of infection when antibody levels were low [33,34]. Another possibility is that the magnitude of cellular and humoral immune responses in each individual could vary. CMV-specific T-cell responses were detected by both ICS and ELISPOT assays in the two individuals classified as anti-CMV IgG seronegative, with relatively high response magnitudes. Importantly, this finding does not necessarily indicate assay false positivity or serological misclassification. Functional T-cell assays such as ELISPOT and ICS are highly sensitive and have been shown to detect CMV-specific memory T-cell responses in individuals lacking detectable serum antibodies, likely reflecting prior subclinical exposure, waning humoral immunity, or immune memory not captured by serology alone [35,36]. The concordant results obtained from two independent functional assays further support the technical validity and biological relevance of these findings. Though these findings suggest that the presence of an antibody response may serve as a predictor of CMIR, this relationship may not appear to be bidirectional. Specifically, the presence of CMIR does not consistently predict a measurable antibody response. Therefore, antibody response should be interpreted with caution when considered as a surrogate marker for CMIR, especially when the antibody response is absent.

Accordingly, this study should be interpreted as a comparative methodological evaluation of two established CMV-specific T-cell assays rather than a diagnostic study aimed at defining CMV serostatus, immune protection, or clinical cut-off values. This study emphasized that humoral immune response could be a surrogate marker for cellular immune response in CMV infection in healthy subjects. This may not extrapolate to immunocompromised patients as the reconstitution between T-cells and antibody-producing B-cells usually do not occur at the same time.

This study has several limitations. The primary limitation was the absence of a clearly defined CMV-seronegative control group, largely due to high CMV seroprevalence in the adult population in our geographic region. This hindered our ability to perform meaningful subgroup analyses based on serostatus [37]. Furthermore, the sample size was relatively small (n = 30), which may limit the precision of the correlation estimates and agreement analyses. Larger studies would be required to confirm the robustness of these findings. Additionally, limitation of this study was that only healthy subjects were enrolled. Testing in the extreme age groups, such as young children and the elderly, and in different groups of patients with varying underlying diseases, especially immunocompromised patients, is warranted. The peptide used in stimulation was only IE1. Using pp65 peptide may yield different results because this peptide tends to induce a stronger CD4$^+$ T-cell response [38]. In addition, this study used only IFN-γ-secreting cells as the surrogate marker for CMV-specific T-cells, which can be detected in the absence of measurable antibodies and that our findings should be interpreted in the context of a methodological comparison rather than diagnostic inference. Cells secreting other cytokines, such as IL-2 and TNF-α, could also be markers for CMV-specific T-cells [39]. Quantifying these cells could improve accuracy but at an increased cost. In addition, the result of the correlation between CMIR and HIR may be different if CMIR was assessed by quantifying cells secreting other cytokines rather than by quantifying only cells secreting IFN-γ. The lack of a universal or standard cut-off value for CMV-specific T-cell responses to determine those with intact immunity is also a limitation. Finally, the percentage of NK cells reported in this study may be underestimated, as cell fixation and permeabilization were performed prior to antibody staining, a process that could alter the recognized epitopes of CD56 and reduce staining efficiency.

## Conclusion

The present study demonstrated that CMV-specific T-cells measured by ICS and ELISPOT assays were well correlated, irrespective of CMV antigens used in stimulation. Additionally, CMV CMIR could be present in the absence of anti-CMV IgG, allowing the detection of cell-mediated immune response in subjects with negative anti-CMV IgG. Both ELISPOT and ICS assays could be the screening and monitoring tools for evaluating the specific T-cell response to CMV infection. ICS would be preferred if quantification of CMV-specific T-cells of lymphocyte subsets is necessary. Anti-CMV IgG level could be a marker of a history of CMV infection and may be used as the surrogate marker for CMIR against CMV infection in immunocompetent hosts.

## Supporting information

**S1 Fig. Gating strategies for all lymphocyte subsets.**
(TIF)

**S1 Table. The percentage of IFN-γ producing cells of individual participants.**
(DOCX)

## Acknowledgments

We would like to thank the individuals who participated in the study. We also thank Nanamon Monnamo, Research Laboratory Section, Office of Health Science Research, Faculty of Medicine Ramathibodi Hospital, Mahidol University, Bangkok, Thailand, for her assistance in collecting peripheral blood samples from healthy participants.

## Author contributions

**Conceptualization:** Chompunuch Klinmalai, Nopporn Apiwattanakul, Somsak Prasongtanakij, Jackrapong Bruminhent, Supanart Srisala.

**Data curation:** Chompunuch Klinmalai, Supanart Srisala.

**Formal analysis:** Chompunuch Klinmalai, Supanart Srisala.

**Funding acquisition:** Supanart Srisala.

**Investigation:** Chompunuch Klinmalai, Nopporn Apiwattanakul, Somsak Prasongtanakij, Supanart Srisala.

**Methodology:** Chompunuch Klinmalai, Somsak Prasongtanakij, Supanart Srisala.

**Supervision:** Nopporn Apiwattanakul, Supanart Srisala.

**Writing – original draft:** Chompunuch Klinmalai, Supanart Srisala.

**Writing – review & editing:** Chompunuch Klinmalai, Nopporn Apiwattanakul, Somsak Prasongtanakij, Jackrapong Bruminhent, Supanart Srisala.

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
