## [Decision Letter · Decision Letter 0]

14 Dec 2025

PONE-D-25-23529Comparison between Enzyme-Linked Immunospot Assay and Intracellular Cytokine Flow Cytometry Assay of Cytomegalovirus-Specific T-Cell Response in Healthy Participants.PLOS One

Dear Dr. Srisala,

Thank you for submitting your manuscript to PLOS ONE. After careful consideration, we feel that it has merit but does not fully meet PLOS ONE’s publication criteria as it currently stands. Therefore, we invite you to submit a revised version of the manuscript that addresses the points raised during the review process.

Overall, the reviewers find the study to be well-conducted, clearly written, and methodologically sound. The comparative analysis of ELISPOT and intracellular cytokine staining assays is considered valuable, particularly in the context of CMV immune monitoring. However, several important issues require clarification and revision.

The major points raised by the reviewers include

Conceptual clarity regarding assay readouts, including precise distinction between cytokine secretion versus intracellular production and the cell-type specificity (or lack thereof) of ELISPOT versus ICS.The interpretation of CMV-seronegative participants who demonstrate CMV-reactive cellular responses raises important implications for data interpretation and study conclusions.Clarification of flow cytometry methodology, including NK cell gating strategy, reported frequencies, and compensation.Handling of background IFN-γ responses, including background subtraction and transparent presentation of unstimulated versus stimulated data.Minor statistical and editorial issues, including missing p-values, participant counts, and justification of variables not central to the study.

We look forward to receiving your revised manuscript.

Kind regards,

Prashant Sharma, Ph.D.

Academic Editor

PLOS One

Journal Requirements:

Faculty of Medicine, Ramathibodi Hospital, Mahidol University, Thailand, Grant Number: RF_66114.

3. Please upload a new copy of Figures 1 and 4 as the detail is not clear. Please follow the link for more information:  https://journals.plos.org/plosone/s/figures

Reviewers' comments:

Reviewer's Responses to Questions

**Comments to the Author**

1. Is the manuscript technically sound, and do the data support the conclusions?

Reviewer #1: Partly

Reviewer #2: Yes

Reviewer #3: Yes

2. Has the statistical analysis been performed appropriately and rigorously? 

Reviewer #1: Yes

Reviewer #2: Yes

Reviewer #3: Yes

3. Have the authors made all data underlying the findings in their manuscript fully available?

Reviewer #1: Yes

Reviewer #2: Yes

Reviewer #3: No

4. Is the manuscript presented in an intelligible fashion and written in standard English?

Reviewer #1: Yes

Reviewer #2: Yes

Reviewer #3: Yes

5. Review Comments to the Author

Reviewer #1: Cell-mediated immune response (CMIR) plays an important role in controlling CMV replication. Monitoring CMIR against CMV is crucial in clinical settings involving immunocompromised patients who are vulnerable to CMV infection or latent reactivation. In this study, the authors compared 2 detection methods of CMIR against CMV in peripheral blood mononuclear cells (PBMCs) from 30 healthy donors: 1). Intracellular staining (ICS) to quantify CMV-specific IFN-g production by T-cell subsets (CD3, CD4, CD8, double negative ND T cells) and 2). Enzyme-linked immunospot (ELISPOT) to quantify CMV-specific IFN-g secreting cells. Both whole CMV lysates and a peptide pool from the CMV IE-I protein were used to stimulate the PBMCs. They found that CMV-specific CD3 T cell responses as measured by ICS correlated well with IFN-g secreting cells quantified by ELISPOT. They also found that the anti-CMV IgG level correlated with the number of IE1-specific IFN-g producing cells.

Overall, the manuscript is well written, and the results generally supported the main conclusions. However, the authors need to be precise in their description of CMV-specific T cell response as measured by ICS and ELISPOT assays. While it is true for ICS, ELISPOT assay is not able to reveal the exact cytokine-producing cell types. Even though IFN-γ-producing cells within the PBMC are mainly CD3+ cells, other cell types such as NK cells can also contribute. In addition, the following issues also need to be addressed/clarified.

The p values for the correlations in the Abstract were missing.

Line 38 in The Abstract stated there were 30 CMV-seropositive participants. Should this be 28?

Table 1: What is relevance of BMI for this study? Does BMI correlate with cytokine production?

Please specify whether the NK-cells were gated on CD3-CD56+ or CD56+ lymphocyte population in the Fig 1 legend and the text. NK cells (CD3-CD56+) usually composed of about 5% to 10% of circulating lymphocytes. Why NK cells represented only 0.2% of total lymphocytes as indicated in Table 2?

Lines 175-177 mixed the function of ICS and ELISPOT assays. ICS, not ELISPOT, quantifies IFN-g-producing T cells.

ICS quantifies cytokine-producing cells, not cytokine-secreting cells. Please be precise in the description in Table 2 and in the text.

Table 3 shows that some subjects such as 9,12, 15 had a high background of IFN-g SFU. Was this background subtracted from the CMV antigen-specific IFN-g SFU? Was this high background IFN-g secretion correlated with high IFN-g production in unstimulated cells by ICS? It would be informative to add a supplemental table to show the % of IFN-g-producing T cells for individual untreated and CMV-stimulated samples.

Fig. S1. CD8 seemed to be under compensated against CD4 in the flow cytometric plot. Please double check flow compensations and show proper gating of NK cells and their cytokine production.

Figure 4A shows the seronegative participant had a high number of SFU, but very few ICS spots. What is the reason for this discrepancy? Which 2 donors were seronegative?

Lines 245-246 stated “Furthermore, a higher significance correlation was found in whole CMV stimulation than in CMV IE-1 peptide pool stimulation.”. Was this based on statistical analysis?

Lines 278-280 stated “Stimulation of cells by peptide pools, IE1 in this study, is mediated by co-stimulation with MHC class I while whole CMV lysate is more likely to stimulate T-cells with

MHC class II.” If IE-1 peptides are HLA class I epitopes, why there were IE-I specific IFN-g response in CD4 cells as well as NK cells? For Table 2, it could be very informative to add % of IFN-g+ unstimulated T cells and NK cells, especially considering the high background of IFN-g SFU.

CD4 cells are composed mainly of CD4, CD8, and DN cells. Why there were lower number of IFN-g+ CD3 cells than IFN-g+ CD4, CD8, or DN?

Reviewer #2: The paper is well written, the methods have been rigorously performed and the results have been discussed critically. The authors have also listed the limitations of their study.

I have no further comments.

Reviewer #3: Review for Klinmalai et al.

Summary

HCMV is a medically relevant pathogen with limited treatment options and very high seroprevalence, so work on its rapid and accurate diagnosis and clinical assessment of patient immune competence is urgently needed. Klinmalai et al. have carried out a quantitative comparison of two of the major methods for determining patient immune responses to HCMV. While the topic they address is not novel, the work appears strong and I would agree that regular re-evaluation of current protocols has value for patient outcomes.

The authors show a comparison of two well-established techniques for assessing immune effector functions: ELISPOT assay, which measures secreted cytokines; and intracellular cytokine staining (ICS), which measures cytokine-containing granules. Both measure only one target (interferon gamma, IFNg) and correlate strongly. While this is the expected outcome – as each of these protocols has been established for some time – it may be valuable to have this confirmatory study added to the body of literature on the topic.

The strength of the work lies in its simplicity, easily quantifiable outcomes, and statistical rigor. A clear positive/negative readout for IFNg-positive cells leaves little doubt as to the validity of each result. The authors use two stimulation techniques, either whole virus lysate or purified peptides, which further strengthen their data.

The paper does contain one very significant weakness, which is the absence of CMV-negative donors. If the authors are able to find CMV-negative donors to act as a negative control population, this work will be strengthened exponentially. Alternatively, given that certain geographic regions have seroprevalence approaching 100% and finding negative donors may be impossible, the authors should adequately justify how they validate their results without such controls.

Overall, I believe this paper has merit and rigor and acts to reinforce our understanding of commonly used diagnostic methods.

Major points.

1. The most important point, as mentioned above, is the lack of appropriate negative controls. Initially, the authors define their cohort as 30 patients, of whom 28 are CMV-seropositive and two -seronegative. However, as they acknowledge, both seronegative patients had a substantial population of HCMV-reactive T cells among their PBMCs. This indicates one of two things: either (A) their seropositivity is incorrect, and they in fact had only HCMV-positive donors in their cohort and no negative controls at all, which weakens the statistical relevance of the study; or (B) the reported seropositivity is correct, and the T cell assays show false-positive results, which weakens the physiological relevance and calls into question the validity of their assay results.

Either of these two cases is problematic for the authors’ conclusions. In case (A), the authors are attempting to publish work lacking adequate controls. This may simply reflect the unavoidable nature of HCMV – in certain demographics and geographic locales, HCMV seropositivity is nearly 100%. I suspect that’s the case in their region, but if so, the authors should adequately address this and explain how it impacts their results.

In case (B), the authors demonstrate nicely that both ELISPOT and ICS are prone to identical false positives. This might indicate a necessity for defining new thresholds positivity or might call into question the value of these methods entirely.

It’s essential that the authors explain to the reader why or why not each of these cases is a valid concern.

Minor points.

1. At line 78, the authors posit that “humoral immune response could be crucial in restricting CMV dissemination.” This statement is, at a minimum, controversial, and requires adequate citation if it is to be included. Many would argue that humoral immunity is neither necessary nor sufficient to limit CMV dissemination.

2. At line 88, the authors mention correlation between anti-CMV IgG and CMIR. Whether they indeed determined its relevance as a surrogate for CMIR is debatable, and given their lack of negative control, might be an overstatement of their findings.

3. Initially, I had some concerns about the presentation of flow cytometry data. However, the supporting document s1 shows a very nice example of how I wish others presented their gating strategies. Very nicely done, and thank you for this transparency.

4. In the discussion, at line 242-243, the authors say that the correlation between ELISPOT and ICS “remains to be elucidated”. Yet, at lines 248-250, they cite at least two studies which do in fact show precisely that correlation. I’m confused, then, what aspect remains to be elucidated.

5. At lines 286-292, the authors make several seemingly contradictory statements about CMIR and humoral immunity: the antibody response can act as a surrogate for CMIR; and yet, at the same time, CMIRs may be functional in the absence of a humoral response. It’s a relatively minor point – because perhaps there truly is value in the correlation, while in a subset of individuals the two metrics simply do not correlate. In its current presentation, it’s a bit confusing.

6. Finally, the authors should be careful in saying that the main limitation was that only healthy subjects were enrolled. It’s true that new information could be found in immunocompromised subjects, but I don’t consider that to be a shortcoming in this particular study. Instead, I’d prefer that they address the issue of no CMV-negative controls.

6. PLOS authors have the option to publish the peer review history of their article (what does this mean?). If published, this will include your full peer review and any attached files.

Reviewer #1: No

Reviewer #2: No

Reviewer #3: No

---

## [Author Response · Author response to Decision Letter 1]

16 Jan 2026

We truly appreciate the editor and the reviewers dedicating to provide constructive feedback. With additional valuable suggestion from the reviewers, we have made efforts to address all the points raised regarding the manuscript. This has greatly improved the clarity of the manuscript and made it more interesting. The changes are also tracking within the manuscript.

For detail of response please see in file "Response to Reviewers"

---

## [Decision Letter · Decision Letter 1]

4 Mar 2026

PONE-D-25-23529R1Comparison between enzyme-linked immunospot assay and intracellular cytokine flow cytometry assay of cytomegalovirus-specific T-cell response in healthy participants.PLOS One

Dear Dr. Srisala,

Thank you for submitting your manuscript to PLOS ONE. After careful consideration, we feel that it has merit but does not fully meet PLOS ONE’s publication criteria as it currently stands. Therefore, we invite you to submit a revised version of the manuscript that addresses the points raised during the review process.

Overall, the reviewers agree that the study is technically sound and clearly presented. The experimental design and statistical analyses are generally appropriate, and the manuscript addresses an important methodological question regarding the comparison of ELISPOT and intracellular cytokine staining (ICS) assays for measuring CMV-specific T-cell responses.

However, substantial revisions are required before the manuscript can be considered further. The principal issues are as follows:

**Analytical Rigor and Agreement Analysis**

The current comparison relies primarily on correlation analysis. Correlation alone does not establish agreement or interchangeability between platforms. Additional analyses (e.g., Bland–Altman plots with limits of agreement and assessment of systematic bias) are necessary to support conclusions regarding assay comparability.

**Positioning of Novelty and Scope**

While prior studies have reported correlations between ELISPOT and ICS assays, the manuscript does not clearly articulate what new knowledge is added. The Discussion should more explicitly define the contribution of this work, particularly in relation to existing literature.

**Interpretation and Clinical Framing**

Given that the study involved healthy, immunocompetent individuals, it is necessary to clearly frame conclusions about clinical monitoring, particularly in immunocompromised populations, as necessitating further validation.

**Methodological Transparency**

Additional technical detail is required for both the flow cytometry and ELISPOT methodologies to ensure reproducibility and compliance with current reporting standards.

**Statistical Clarifications and Presentation**

Please provide confidence intervals for correlation coefficients and address sample size limitations explicitly.

We believe that these concerns are addressable without additional experimental data, but they do require substantial analytical and interpretive strengthening.

We look forward to receiving your revised manuscript.

Kind regards,

Prashant Sharma, Ph.D.

Academic Editor

PLOS One

Journal Requirements:

Reviewers' comments:

Reviewer's Responses to Questions

**Comments to the Author**

1. If the authors have adequately addressed your comments raised in a previous round of review and you feel that this manuscript is now acceptable for publication, you may indicate that here to bypass the “Comments to the Author” section, enter your conflict of interest statement in the “Confidential to Editor” section, and submit your "Accept" recommendation.

Reviewer #3: All comments have been addressed

Reviewer #4: (No Response)

Reviewer #5: (No Response)

Reviewer #6: (No Response)

2. Is the manuscript technically sound, and do the data support the conclusions?

Reviewer #3: Yes

Reviewer #4: No

Reviewer #5: Yes

Reviewer #6: Partly

3. Has the statistical analysis been performed appropriately and rigorously? 

Reviewer #3: Yes

Reviewer #4: I Don't Know

Reviewer #5: Yes

Reviewer #6: No

4. Have the authors made all data underlying the findings in their manuscript fully available?

Reviewer #3: No

Reviewer #4: No

Reviewer #5: (No Response)

Reviewer #6: Yes

5. Is the manuscript presented in an intelligible fashion and written in standard English?

Reviewer #3: Yes

Reviewer #4: No

Reviewer #5: Yes

Reviewer #6: Yes

6. Review Comments to the Author

Reviewer #3: Thank you for your willingness to address all of my comments. I think the manuscript is substantially improved by your clarifications and I have no reservations in recommending for publication.

Reviewer #4: This article describes the humoral and cellular immune responses to CMV antigens in 30 healthy subjects. The study reports a correlation between the ELISPOT assay and intracellular IFN production in T lymphocytes, as measured by flow cytometry. In this respect, the study does not highlight any novel findings, as both methods assess cellular immune responses and a correlation between them is expected. It would be more interesting if the study were extended to include and compare immunosuppressed patients, to determine whether differences are observed in the patient population.

In Abstract: Lines 37, 38, and 39 contain unnecessary repetition about the seropositivity of the study participants.

The last sentence in lines 42–43: “CMV IgG level could be a marker of past CMV infection and could be correlated with CMIR against CMV infection in immunocompetent hosts” is unclear in its meaning.

In Introduction: Lines 49–50: The statement “Reactivation of CMV in healthy individuals is usually asymptomatic” is unclear and contradictory in wording.

Lines 54–55: The sentence “Interferon-gamma (IFN-γ) secreted by these cells upon stimulation with specific antigens could represent that these cells have been activated [3–6]” is incorrect.

Lines 58–59: The sentence “Nevertheless, other immunologically active cells also play a role in the defense against CMV infection” is also unclear, as it does not specify which cells are being referred to and lacks a reference; the subsequent sentence should be expressed more clearly.

Lines 81–82: The sentence “Humoral immune response (HIR) could be crucial in restricting CMV dissemination and minimizing disease severity” requires a reference.

Lines 86–88: The sentence “However, anti-CMV IgG may not provide overall immunological status against CMV infection, especially regarding CMIR” is inaccurate and requires a reference.

In Methods: The authors should describe how the individuals included in the study were selected.

Reviewer #5: This manuscript presents a technically sound comparison of ELISPOT and intracellular cytokine staining (ICS) assays for measuring CMV-specific T-cell responses in healthy individuals. The study design, methodology, and statistical analyses are appropriate, and the conclusions are supported by the data. The authors have appropriately acknowledged the study limitations and provided a reasonable interpretation of CMV-specific T-cell responses in seronegative individuals. Overall, the manuscript is well written and provides useful insight for CMV immune monitoring. I have no major concerns.

Reviewer #6: Summary

This manuscript compares intracellular cytokine staining (ICS) flow cytometry and IFN-γ ELISPOT assays for quantifying CMV-specific T-cell responses in healthy adults. The authors report moderate to strong correlations between the two platforms and explore associations with anti-CMV IgG levels. The study is technically well executed and clearly presented. However, the current manuscript is primarily confirmatory and requires strengthening in analytical rigor, positioning of novelty, and interpretation of clinical relevance.

With substantial revision, the manuscript could be suitable for publication.

Major Comments

1. Novelty and scientific positioning need strengthening

While the study is carefully performed, the central finding correlation between ELISPOT and ICS has been demonstrated previously. The authors themselves acknowledge prior reports of significant correlations between these assays.

Concerns

• The manuscript does not clearly articulate what new knowledge is added.

• It is unclear how this work advances CMV immune monitoring beyond existing literature.

• The translational implication remains underdeveloped.

Recommendations

The authors should explicitly clarify:

• What gap in the literature this study fills

• Whether this represents analytical validation in a specific population

• How the findings would influence assay selection in practice

The Discussion should more clearly differentiate this work from prior comparisons.

2. Cohort selection limits clinical relevance

The study is conducted entirely in healthy immunocompetent adults (n = 30), yet the clinical motivation repeatedly emphasizes use in immunocompromised patients (e.g., transplant recipients).

Concerns

• Immune dynamics in healthy adults may not reflect the intended clinical population.

• The conclusions regarding monitoring utility may therefore be overstated.

• The very high CMV seroprevalence (93.3%) further limits discriminatory power.

Recommendations

Please:

• Provide a stronger rationale for using a healthy cohort for assay comparison.

• Temper clinical claims accordingly.

• Clearly state that validation in immunocompromised populations is required.

3. Method comparison is incomplete (correlation alone is insufficient)

The study relies primarily on Spearman correlation to compare assays. However, correlation does not establish agreement or interchangeability.

Why this matters

Two assays can correlate well while still showing clinically meaningful bias.

Required analyses

The authors should add:

• Bland Altman analysis (strongly recommended)

• Assessment of systematic bias

• Limits of agreement

• Ideally Deming or Passing Bablok regression

Without agreement analysis, the conclusion that the assays could be used interchangeably is not fully supported.

4. Sample size and statistical power

The sample size is small (n = 30), and no power calculation is provided.

Concerns

• Stability of correlation estimates is uncertain.

• Confidence intervals for correlation coefficients are not reported.

• The high seropositivity limits meaningful subgroup analysis.

Recommendations

Please:

• Provide a sample size justification or post hoc power estimate.

• Report 95% confidence intervals for correlation coefficients.

• Discuss the limitations imposed by the small sample size more explicitly.

5. Interpretation of seronegative but T-cell–positive individuals

The finding that two seronegative individuals had high CMV-specific T cells is interesting and potentially important.

However, the interpretation is currently somewhat one-sided.

Concerns

The discussion focuses mainly on biological explanations (e.g., prior exposure, waning antibodies) but does not sufficiently consider:

• assay background or false positivity

• cross-reactive T-cell responses

• threshold definition

• technical variability

Recommendations

Please expand the discussion to include alternative explanations and provide supporting literature where possible. This observation could be a strength of the paper if interpreted more rigorously.

6. Flow cytometry reporting requires additional detail

The ICS methodology would benefit from more comprehensive reporting to meet current flow cytometry standards.

Please clarify or add

• Doublet exclusion strategy

• Viability dye usage (if any)

• Compensation controls

• Minimum events acquired

• Instrument QC procedures

• Representative gating hierarchy in main text (not only supplement)

Given the technical nature of the comparison, more detailed reporting is important for reproducibility.

7. ELISPOT methodological detail

Several technical aspects of the ELISPOT assay are insufficiently described.

Please clarify

• Number of replicate wells per condition

• Criteria for spot size and counting thresholds

• Handling of high spot counts (>1000/well mentioned but not quantified)

• Intra-assay variability or coefficient of variation

These details are important for evaluating assay robustness.

8. Conclusions are somewhat overstated

The manuscript concludes that both assays “could be the screening and monitoring tools” for CMV-specific T-cell responses.

Given:

• small healthy cohort

• absence of clinical outcomes

• lack of agreement analysis

this statement should be moderated.

Recommendation

Rephrase to emphasize analytical correlation and the need for further validation in clinical populations.

Minor Comments

1. NK cell frequency appears unusually low

The reported median NK cell proportion (~0.2%) is substantially below expected physiological ranges.

Although the authors note possible underestimation, this issue deserves more prominent discussion and potential methodological clarification.

2. Blood processing and cell viability

Samples were processed within 3 hours at room temperature.

Please report:

• PBMC viability after isolation

• whether viability thresholds were applied

3. Statistical methods clarity

Please clarify:

• which comparisons used paired vs unpaired tests

• whether normality was assessed

• whether multiple testing correction was considered

4. Anti-CMV IgG handling

The manuscript states that values >250 AU/mL were truncated for analysis.

Please justify this approach and discuss its potential impact on correlation estimates.

5. Figure presentation

Consider improving clarity of Figure 3 by:

• adding regression lines with confidence intervals

• harmonizing axis scales

• considering log transformation if data are skewed

6. Language and style

The manuscript is generally clear but would benefit from minor English editing to improve flow and remove some redundancy in the Introduction and Discussion.

Overall Recommendation

Major revision

The study is technically sound and addresses a relevant methodological question. However, additional analytical rigor, clearer positioning of novelty, and more cautious interpretation are required before the manuscript is suitable for publication.

7. PLOS authors have the option to publish the peer review history of their article (what does this mean?). If published, this will include your full peer review and any attached files.

Reviewer #3: **Yes:** Timothy M. White

Reviewer #4: No

Reviewer #5: **Yes:** Lawrence Annison

Reviewer #6: **Yes:** ANURAG ADHIKARI

---

## [Author Response · Author response to Decision Letter 2]

2 Apr 2026

We would like to sincerely thank the Academic Editor and the reviewers for their careful evaluation of our manuscript and for their constructive comments, which helped improve the quality and clarity of our work. We have carefully revised the manuscript in accordance with all suggestions. All changes have been incorporated into the revised manuscript and are highlighted in the tracked version.

---

## [Decision Letter · Decision Letter 2]

29 Apr 2026

Comparison between enzyme-linked immunospot assay and intracellular cytokine flow cytometry assay of cytomegalovirus-specific T-cell response in healthy participants.

PONE-D-25-23529R2

Dear Dr. Srisala,

We’re pleased to inform you that your manuscript has been judged scientifically suitable for publication and will be formally accepted for publication once it meets all outstanding technical requirements.

Kind regards,

Prashant Sharma, Ph.D.

Academic Editor

PLOS One

Additional Editor Comments (optional):

Reviewers' comments:

Reviewer's Responses to Questions

**Comments to the Author**

1. If the authors have adequately addressed your comments raised in a previous round of review and you feel that this manuscript is now acceptable for publication, you may indicate that here to bypass the “Comments to the Author” section, enter your conflict of interest statement in the “Confidential to Editor” section, and submit your "Accept" recommendation.

Reviewer #3: All comments have been addressed

Reviewer #4: All comments have been addressed

2. Is the manuscript technically sound, and do the data support the conclusions?

Reviewer #3: Yes

Reviewer #4: Yes

3. Has the statistical analysis been performed appropriately and rigorously? 

Reviewer #3: Yes

Reviewer #4: Yes

4. Have the authors made all data underlying the findings in their manuscript fully available?

Reviewer #3: Yes

Reviewer #4: Yes

5. Is the manuscript presented in an intelligible fashion and written in standard English?

Reviewer #3: Yes

Reviewer #4: Yes

6. Review Comments to the Author

Reviewer #3: Recent revisions have continued to improve the clarity and quality of language in the manuscript, for which I appreciate the authors' diligence. I believe it should be accepted.

Reviewer #4: The authors have made all the necessary changes requested by the reviewers. The article is now more complete and presents the objectives of the work in a more convincing manner. I believe that in this form, the article can be accepted for publication.

7. PLOS authors have the option to publish the peer review history of their article (what does this mean?). If published, this will include your full peer review and any attached files.

Reviewer #3: No

Reviewer #4: No

---

## [Editor Report · Acceptance letter]

PONE-D-25-23529R2

PLOS One

Dear Dr. Srisala,

I'm pleased to inform you that your manuscript has been deemed suitable for publication in PLOS One. Congratulations! Your manuscript is now being handed over to our production team.

Kind regards,

on behalf of

Dr. Prashant Sharma

Academic Editor

PLOS One